# Multidrug-Resistant and Extensively Drug-Resistant *Acinetobacter baumannii* Causing Nosocomial Meningitis in the Neurological Intensive Care Unit

**DOI:** 10.3390/microorganisms11082020

**Published:** 2023-08-06

**Authors:** Nadezhda K. Fursova, Mikhail V. Fursov, Evgeny I. Astashkin, Anastasiia D. Fursova, Tatiana S. Novikova, Angelina A. Kislichkina, Angelika A. Sizova, Galina N. Fedyukina, Ivan A. Savin, Olga N. Ershova

**Affiliations:** 1Department of Molecular Microbiology, State Research Center for Applied Microbiology and Biotechnology, Territory “Kvartal A”, 142279 Obolensk, Russia; ast.ev@mail.ru (E.I.A.); anfursova06@gmail.com (A.D.F.); pozitifka.15@yandex.ru (T.S.N.); 2Department of Training and Improvement of Specialists, State Research Center for Applied Microbiology and Biotechnology, Territory “Kvartal A”, 142279 Obolensk, Russia; mikhail.fursov88@gmail.com; 3Department of Culture Collection, State Research Center for Applied Microbiology and Biotechnology, Territory “Kvartal A”, 142279 Obolensk, Russia; angelinakislichkina@yandex.ru (A.A.K.); sizova1508@gmail.com (A.A.S.); 4Department of Immunochemistry of Pathogenic Microorganisms, State Research Center for Applied Microbiology and Biotechnology, Territory “Kvartal A”, 142279 Obolensk, Russia; galafed@mail.ru; 5Department of Clinical Epidemiology, National Medical Research Center of Neurosurgery Named after Academician N.N. Burdenko, 125047 Moscow, Russia; savin@nsi.ru (I.A.S.); oershova@nsi.ru (O.N.E.)

**Keywords:** *Acinetobacter baumannii*, meningitis, ICU, MDR, XDR, virulence genes, biofilm

## Abstract

*Acinetobacter baumannii* is one of the significant healthcare-associated meningitis agents characterized by multidrug resistance and a high mortality risk. Thirty-seven *A. baumannii* strains were isolated from thirty-seven patients of Moscow neuro-ICU with meningitis in 2013–2020. The death rate was 37.8%. Strain susceptibility to antimicrobials was determined on the Vitek-2 instrument. Whole-genome sequencing was conducted using Illumina technology; the sequence types (ST), capsular types (KL), lipooligosaccharide outer core locus (OCL), antimicrobial resistance genes, and virulence genes were identified. The prevalent ST was ST2, belonging to the international clone IC2, and rarer, ST1, ST19, ST45, ST78, ST106, and ST400, with prevalence of KL9 and OCL1. Twenty-nine strains belonged to multidrug-resistant (MDR) and eight extensively drug-resistant (XDR) categories. Genes conferring resistance to beta-lactams (*bla*_PER_, *bla*_GES_, *bla*_ADC_, *bla*_CARB_, *bla*_CTX-M_, *bla*_TEM_, and *bla*_OXA_-types), aminoglycosides (*aac*, *aad*, *ant*, *aph*, and *arm*), tetracyclines (*tet*), macrolides (*msr* and *mph*), phenicols (*cml*, *cat*, and *flo*), sulfonamides (*dfr* and *sul*), rifampin (*arr*), and antiseptics (*qac*) were identified. Virulence genes of nine groups (Adherence, Biofilm formation, Enzymes, Immune evasion, Iron uptake, Regulation, Serum resistance, Stress adaptation, and Antiphagocytosis) were detected. The study highlights the heterogeneity in genetic clones, antimicrobial resistance, and virulence genes variability among the agents of *A. baumannii* meningitis, with the prevalence of the dominant international clone IC2.

## 1. Introduction

Carbapenem-resistant *Acinetobacter baumannii* (CRAB) has recently been labeled as a “critical” pathogen, i.e., constituting a significant global human health risk, in the World Health Organization’s list of antimicrobial-resistant bacteria intended to guide the research and development of new effective antibiotic treatments [1]. The recent COVID-19 pandemic was associated with an increasing risk of hospital-acquired ventilator-associated secondary infections, for which *A. baumannii* was a leading cause [2]. *A. baumannii* has been frequently reported as the causative pathogen of nosocomial infections, such as ventilator-associated pneumonia, bloodstream infection, urinary tract infection, wound infection, and meningitis. Nosocomial meningitis is a potentially life-threatening central nervous system infection associated with increased mortality, morbidity, and hospital costs; *A. baumannii* accounted for more than 25% of all pathogens isolated from cerebrospinal fluid (CSF) [3]. The increasing prevalence of CRAB has become a serious clinical problem in many countries because the mortality rate of CRAB meningitis has been reported to exceed 70% [4,5].

The predominance of a few successful multidrug-resistant lineages worldwide underlines the importance of elucidating the epidemiology of *A. baumannii* isolates in single hospitals, at a country-wide level, and on a global scale. The population structure of *A. baumannii* is composed of *A. baumannii* strains assigned to 176 different sequence types (STs) that can be grouped into 20 clonal complexes (CCs) according to Pasteur’s Multilocus sequence typing (MLST) scheme [6]. Three *A. baumannii* international clones (ICs) have been determined as prevalent: IC I (includes ST1, ST7, ST8, ST19, and ST20); IC II (includes ST2, ST45, ST47, and ST59); IC III (includes ST3, ST13, and ST14). In Europe, IC I and IC II cause the vast majority of infections, and IC II was reported in most cases. IC I, IC II, and IC III were characterized by increasing resistance to clinically significant antimicrobials [7,8]. The emergent IC VI (ST78) strains belonged to the CRAB and presented high biofilm-forming ability, increased resistance to drying, and an enhanced capacity for host cell adhesion/invasion, favoring its diffusion and persistence, and have been identified in European countries (Italy, Russia, Greece, and Germany), Asia (Kuwait), and North and South America (the USA and French Guiana) [9,10,11]. Moreover, the new sequence type ST400 is not related to those of the main clonal complexes known to have spread worldwide [12].

*A. baumannii* carbapenem resistance is mainly associated with the production of carbapenemases, and more rarely, with the efflux pumps and with the reduction or inactivation of porins expression [13]. The most common *A. baumannii* carbapenemases are OXA-23-like, OXA-24/40-like, and OXA-58-like enzymes [14]. The prevalence of OXA-23-like-producers is attributed to the spread of successful global clones such as GC1 and GC2 [15]. Moreover, *A. baumannii* resistomes contain other groups of beta-lactamases, e.g., the extended-spectrum beta-lactamases (ESBLs) SHV-5, TEM-92, CTX-M-2, CTX-M-15, PER-1, PER-2, PER-7, VEB-1, and GES-14, narrow-spectrum beta-lactamases (NSBLs) TEM-1 and SCO-1, and AmpC beta-lactamases ADC-30, ADC-56, etc. [16]. *A. baumannii* is a life-threatening problem not only because of multidrug resistance but also its ability to evade the host immune response, to survive under stressful environmental conditions, its resistance to disinfectants, and its ability to form complex structures of biofilms. It was shown that there are several master regulators of antimicrobial resistance, stress response, and virulence in *A. baumannii*, such as GigAB and DksA [17,18].

The virulence factors of *A. baumannii* cells include (i) the adherence due to Acinetobacter trimeric autotransporter Ata and Type IV pili TFP; (ii) the effector delivery system including type II and VI secretion systems T2SS and T6SS, as well as T6SS effectors Tse1, Tse2, Tse3, and Tse4; (iii) the exotoxins, e.g., phospholipases C and D; (iv) the exoenzyme, coagulation targeting metalloendopeptidase CpaA; (v) the immune modulation due to capsule, lipopolysaccharide LPS, outer membrane protein A OmpA, and penicillin-binding protein PbpG; (vi) the biofilm formation associated with efflux pump AdeFGH, biofilm-associated protein Bap, fimbriae Csu, poly-(1,6)-N-acetyl glucosamine PNAG, and the autoinducer–receptor mechanism for quorum sensing; and (vii) the nutritional/metabolic factors such as acinetobactin and Heme uptake system [19].

This study analyzed the antimicrobial susceptibility, biofilm formation, and virulence factors, as well as their genetic determinants in whole-genome sequences, of 37 *A. baumannii* strains that caused meningitis in 37 patients (mortality rate 37.8%) in Moscow Neurological intensive care unit in 2013 to 2020. Seven sequence types were identified in this study using the Pasteur MLST scheme: ST1, ST2, ST19, ST45, ST78, ST106, and ST400. The most prevalent (23/37) was ST2, which was associated with 12 of 14 patient deaths. The results are significant for expanding our knowledge about the molecular epidemiology of the nosocomial *A. baumannii* in critical care units in Russia. The resistome and virulome analysis provides the data that can be used for the development of control strategies against *A. baumannii* infections and novel antimicrobial and anti-virulence targets for detection systems and treatment.

## 2. Materials and Methods

### 2.1. Research, Bioethical Requirements, and Patients

This study was a retrospective cohort observation analytical study. Under the requirements of the Russian Federation Bioethical Committee, each patient signed informed voluntary consent to treatment and laboratory examination. The study did not contain the personal data of patients, such as the name, date of birth, address, and disease history. The study was a retrospective observational study in the neuro-ICU department in a specialized Neurosurgical Hospital in Moscow, Russia, with 300 beds that care for approximately 8000 patients per year, 95% of whom undergo surgery [20].

### 2.2. Bacterial Isolates and Growing

*A. baumannii* isolates (n = 37) were collected from clinical samples (cerebrospinal fluid) of the patients (n = 37) in the neuro-ICU in the period from January 2013 to September 2020. Bacterial identification was performed using a MALDI-TOF Biotyper (Bruker Daltonics, Bremen, Germany) instrument. Bacterial isolates were grown at 37 °C on Nutrient Medium No. 1 (SRCAMB, Obolensk, Moscow region, Russia), Luria–Bertani broth (Difco Laboratories, Detroit, MI, USA), and Muller–Hinton broth (Himedia, Mumbai, Maharashtra, India). Bacterial isolates were stored in 15% glycerol at minus 80 °C.

### 2.3. Antimicrobial Susceptibility

Susceptibility to 23 antimicrobials (AMs) of 11 functional groups—aminoglycosides (gentamicin, tobramycin, amikacin, netilmicin); carbapenems (imipenem, meropenem), penicillin (ampicillin); penicillin/beta-lactam inhibitors (ampicillin–sulbactam, piperacillin/tazobactam); cephalosporins (cefuroxime, cefoxitin, cefotaxime, ceftazidime, ceftriaxone, cefepime, cefoperazone/sulbactam); fluoroquinolones (ciprofloxacin); sulfonamides (trimethoprim–sulfamethoxazole); phenicols (chloramphenicol); tetracyclines (tetracycline, tigecycline); nitrofurans (nitrofurantoin); and polymyxins (colistin)—were determined using Vitek-2 Compact instrument (BioMerieux, Paris, France). The results were interpreted according to the European Committee on Antimicrobial Susceptibility Testing, Version 13.0, 2023 (http://www.eucast.org, access date: 21 March 2023). Reference strains *Escherichia coli* ATCC 25922 and ATCC 35218 were used as quality controls. The isolates were categorized as multidrug-resistant (MDR), non-susceptible to ≥1 agent in ≥3 antimicrobial groups; and extensively drug-resistant (XDR), non-susceptible to ≥1 agent in all but ≤2 groups according to the criteria proposed by Magiorakos et al. [21].

Minimal inhibitory concentrations (MICs) of two antiseptics—chlorhexidine digluconate (Sigma-Aldrich, Saint Louis, MO, USA) and benzalkonium chloride (Sigma-Aldrich, Saint Louis, MO, USA)—were determined via the broth microdilution assay on the 96-well plates. Bacterial growth in the wells was detected via the xMARK microplate spectrophotometer (Bio-Rad Laboratories, Inc., Watford, UK). MIC values were compared with antiseptic concentrations recommended for use in ICU: 500 mg/L chlorhexidine digluconate; 10,000 mg/L benzalkonium chloride [22].

### 2.4. Biofilm Formation

The biofilm formation ability of *A. baumannii* isolates was determined via polystyrene tube assay based on the crystal violet staining method [23]. Briefly, the wells of a 96-well plate containing 0.1 mL of Luria broth (Thermo Fisher Scientific, Waltham, MA, USA) were inoculated with 100 µL of overnight bacterial culture in concentration 1 × 10^7^ CFU/mL and incubated at 37 °C for ~20 h. The liquid media was discarded, and the adherent cells were washed twice with 0.2 mL of phosphate-buffered saline (PBS) and stained with 0.1% solution of crystal violet for 10 min. The stain was discarded, and the cells were washed three times with 0.2 mL of PBS. The stain was eluted from the adherent cells via 0.2 mL of 96% ethanol for 5 min and transferred in. The absorbance of the eluted stain was measured at 600 nm using the xMark microplate spectrophotometer (Bio-Rad, Hercules, CA, USA). The results were divided into four categories according to the sample optical densities (ODs) in comparison with the control optical densities (ODc): strong biofilm producer (4 × ODc < ODs); medium biofilm producer (2 × ODc < ODs ≤ 4 × ODc); weak biofilm producer (ODc < ODs ≤ 2 × ODc); and non-biofilm producer (ODs ≤ ODc [24].

### 2.5. Whole-Genome Sequencing, Assembly, and Annotation

DNA isolation was performed via the CTAB method [25]. WGS was carried out using Nextera DNA Library Preparation Kit (Illumina, San Diego, CA, USA) and MiSeq Reagent Kits v3 (Illumina, San Diego, CA, USA) for platform Illumina MiSeq (Illumina, San Diego, CA, USA). Moreover, WGS was performed on MGI (MGI Tech, Shenzhen, China) platform using MGIEasy FS DNA Library Prep Kit MGI-Seq (MGI Tech, Shenzhen, China) and 2000RS High-throughput sequencing kit PE200 (MGI Tech, Shenzhen, China). The assemblies of the genomes were obtained using Unicycler v. 0.4.7 software (The University of Melbourne, Victoria, Australia) with default settings that included primary filtering and quality control [26]. Annotation was carried out via NCBI Prokaryotic Genome Annotation Pipeline (PGAP) v. 6.4 and v. 6.5 (National Center for Biotechnology Information, Bethesda, MD, USA) [27].

### 2.6. Multilocus Sequence Typing, KL-Typing, and OCL-Typing

Sequence Types were identified using the MLST resource of the Center for Genomic Epidemiology (CGE) with a 95% threshold for minimum identity and 60% minimum coverage (http://www.genomicepidemiology.org/, access date: 20 March 2023). *A. baumannii* capsular types (KL) and lipooligosaccharide outer core locus (OCL) were identified using Kaptive software v.2.0.7 (https://www.microbiology research.org/content/journal/mgen/, access date: 30 March 2023) [28].

### 2.7. Antimicrobial Resistance and Virulence Genes

Antimicrobial resistance genes were recognized using ResFinder and KmerResistance resources of CGE with a 90% identity threshold and 10% threshold for depth corr. (http://www.genomicepidemiology.org/, access date: 20 March 2023). The virulence factor database (VFDB) online resource was used to identify virulence genes within the genomes with e-value cut-off 1 × 10^−5^, identity > 60%, and coverage > 90% (http://www.mgc.ac.cn/VFs/, access date: 20 March 2023) [29].

### 2.8. Phylogenetic Analysis

The phylogenetic tree of *A. baumannii* whole genomes was constructed with core SNPs identified via WOMBAC (https://github.com/tseemann/wombac, access date: 20 March 2023) (https://github.com/tseemann/wombac, access date: 20 March 2023) and visualized with SplitsTree4 (https://github.com/husonlab/splitstree4, access date: 20 March 2023) using the neighbor-joining (NJ) method [30].

### 2.9. Nucleotide Sequences Accession Numbers

The complete sequences of 37 isolates were deposited under BioProject number PRJNA269675: JAROBQ000000000, JASKJB000000000, JASKJA000000000, JAROBM000000000, JAROBU000000000, JASKIZ000000000, JAROBS000000000, JAROBO000000000, JAROBG000000000, JASKIY000000000, JASKIX000000000, JASKIW000000000, JASKIV000000000, JASKIU000000000, JAROBR000000000, JAROBT000000000, JASKIT000000000, JAROBB000000000, JAROBI000000000, JASKIR000000000, JASKIQ000000000, JASKIP000000000, JASKIO000000000, JASKIN000000000, JASKIM000000000, JASKIL000000000, JASKIK000000000, JASKIJ000000000, JASKII000000000, JASKIH000000000, JASKIG000000000, JASKIF000000000, JASKIE000000000, JASKID000000000, JASKIC000000000, JASKIB000000000.

## 3. Results

### 3.1. Patients and Clinical Data

During the period from January 2013 to Sepember 2020, 37 cases of *A. baumannii* meningitis (MG) were detected in a Moscow neurosurgery ICU in the patients after neurosurgery operations. Among them were 25 males of 12–75 years (median 46.7) and 12 females of 2–65 years (median 44.8). The diagnoses of the patients were oncological (n = 25), traumatic brain injury (n = 8), and disorders of the cerebral circulation including ruptured aneurysms (n = 4). Meningitis was detected in 37 patients; moreover, 19 respiratory tract infections (RTIs), 7 surgical site infections (SSIs), 5 gastrointestinal dysfunctions (GD), 5 urinary tract infections (UTIs), and 1 blood infection (BI) were detected in combinations from one (MG) to four (MG + RTI + SSI + GD or MG + RTI + UTI + GD) infections. Infections were located in the brain and other body sites simultaneously. The stays in the ICU were 8–229 days (median 67.6 days). The death rate was estimated as 37.8% (14/37 patients) (Table 1).

### 3.2. Sequence Types, Capsular Polysaccharide Locus Types, and Lipooligosaccharide Outer Core Locus Types

Sequence Type (ST), Capsular Polysaccharide Locus (KL) Type, and Lipooligosaccharide Outer Core Locus (OCL) Type were identified using whole genomes of *A. baumannii* strains. Seven STs were revealed: ST1 (n = 3); ST2 (n = 15); ST19 (n = 3); ST45 (n = 2); ST78 (n = 2); ST106 (n = 1); and ST400 (n = 1). The prevalent ST2 belonging to the pandemic lineage, named “international clone IC2”, was associated in our study with 12 deaths. Thirteen KLs were identified. The prevalent KL was KL9 (n = 14), following KL2 (n = 3), KL3 (n = 3), KL4 (n = 2), KL15 (n = 2), KL17 (n = 1), KL49 (n = 2), KL77 (n = 1), KL91 (n = 3), KL104 (n = 1), KL165 (n = 1), KL213 (n = 2), and KL235 (n = 2). The major OCL was OCL1 (n = 33), rarer–OCL5 (n = 3), and OCL6 (n = 1) (Table 2).

### 3.3. Antimicrobial Susceptibility

All *A. baumannii* strains in this study were attributed to two antimicrobial resistance patterns: MDR (n = 29) and XDR (n = 8), according to Magiorakos et al.’s criterium [21]. Major strains were resistant to penicillin and cephalosporins (n = 37), fluoroquinolones (n = 36), aminoglycosides (n = 32), penicillin/beta-lactam inhibitors (n = 28), sulfonamides (n = 26), carbapenems (n = 25), and chloramphenicol (n = 20). At the same time, dominant strains were susceptible to polymyxins (n = 34) and tetracyclines (n = 26) (Figure 1).

Nine strains were resistant to six antimicrobial groups, seven to seven groups, fifteen to eight groups, four to nine groups, and two to ten antimicrobial groups simultaneously. The phenotypic diversity of antibiotic resistance among the strains was very high—23 variants of AMR phenotype were revealed. Carbapenem-resistant strains represented ~68% of strains (Table 2).

It was shown that planktonic cells of all *A. baumannii* strains were sensitive to chlorhexidine digluconate and benzalkonium chloride, with MICs 8–64 mg/L and 4–8 mg/L, respectively. Such levels of susceptibility to antiseptics are within their concentrations recommended for use in ICUs (http://dezreestr.ru/, access date 30 March 2023): 500 mg/L for chlorhexidine digluconate; 10,000 mg/L for benzalkonium chloride (Table 3).

### 3.4. Biofilm Formation

*A. baumannii* strains were divided into four groups according to their ability for biofilm (BF) formation: strong BF (n = 2); moderate BF (n = 8); weak BF (n = 17); and no BF (n = 10). Notably, strong BF formation was detected for the strains of ST2 only, and moderate BF formation for the strains of ST2, ST1, and ST400 (Figure 2).

### 3.5. Whole-Genome Analysis

The complete genome assemblies of 37 *A. baumannii* strains contained 3553–3891 coding genes, 48–108 pseudogenes, and 69–71 RNA genes. The GC content of the genomes varied from 39.1 to 42.9% (Table 4).

### 3.6. The Resistome Identification and Analysis

The resistome analysis of 37 *A. baumannii* genomes showed the presence of a total of 47 types of antimicrobial resistance genes (ARGs) providing resistance to six functional groups of antimicrobials, namely, resistance to aminoglycosides (n = 16), beta-lactams (n = 17), tetracyclines (n = 1), macrolides (n = 2), phenicols (n = 4), sulfonamides (n = 3), rifamycin (n = 1), and antiseptics (n = 1).

Aminoglycoside resistance was associated with aminoglycoside N-acetyltransferase *aac* (n = 32), aminoglycoside 6-adenylyltransferase *aad* (n = 23), aminoglycoside nucleotidyltransferase *ant* (n = 5), aminoglycoside O-phosphotransferase *aph* (n = 63), and 16S rRNA (guanine(1405)-N(7))-methyltransferase *arm* (n = 12) genes. The strains of ST1 carried 3–6 ARGs of this group, the strains of ST2 carried 3–7 ARGs each, the strains of ST1 carried 9–2 ARGs each, the strains of ST4 carried 5–3 ARGs each, the strains of ST7 and 8 carried 1–3 ARGs each, the strain of ST400 carried 6 ARGs, and the strain of ST106 had no aminoglycoside resistance genes.

Beta-lactamase ARGs were detected as follows: broad-spectrum class A beta-lactamase *bla*_TEM-1_ gene (n = 6), carbenicillin-hydrolyzing class A beta-lactamase *bla*_CARB-14_ gene (n = 2), extended-spectrum beta-lactamase (ESBL) of class A *bla*_CTX-M-124_ (n = 3), inhibitor-resistant ESBL of class A *bla*_PER-1_ (n = 7), and ESBL of class *bla*_PER-7_ (n = 2), *bla*_GES-11_ (n = 1), and *bla*_GES-12_ (n = 2). *Acinetobacter*-specific ESBL of class C *bla*_ADC-25_ gene was detected in all 37 strains. Notably, beta-lactamase genes *bla*_TEM-1B_, *bla*_CARB-14_, and *bla*_CTX-M-124_ were identified in the genomes of the strains attributed to ST78 only, while *bla*_TEM-1D_ gene was identified in the strains of ST2. The *bla*_PER_ genes were detected in the strains belonging to ST2, ST45, and ST400, whereas *bla*_GES_ genes were detected in the strains of ST1 and ST400. Carbapenem resistance of the studied strains was supported by eight OXA-type carbapenemase genes: *bla*_OXA-23_ (n = 5); *bla*_OXA-72_ (n = 19); *bla*_OXA-66_ (n = 23); *bla*_OXA-69_ (n = 6); *bla*_OXA-80_ (n = 3); *bla*_OXA-90_ (n = 3); *bla*_OXA-100_ (n = 1); and *bla*_OXA-106_ (n = 1). Interestingly that *bla*_OXA-23_ and *bla*_OXA-80_ carbapenemase genes were detected in the strains belonging to ST2 only; *bla*_OXA-66_ gene in the strains of ST2 and ST45; *bla*_OXA-69_ gene in the strains of ST1 and ST19; *bla*_OXA-90_ gene in the strains of ST78; and *bla*_OXA-100_ and *bla*_OXA-106_ genes—in the stains of ST400 and ST106, respectively.

The *tet* gene of tetracycline efflux MFS-family transporter was detected in 1 strain of ST1 and 18 strains of ST2. The *msrE* and *mphE* genes coding ribosomal protection protein and macrolide 2′-phosphotransferase, respectively, were simultaneously identified in the strains of ST2 (n = 11) and ST78 (n = 1). Regarding phenicol resistance markers, the *cat* (n = 9) genes encoded subunits of chloramphenicol acetyltransferase, and the *cmlA* (n = 6) and *floR* (n = 2) genes encoded chloramphenicol/florfenicol efflux MFS-family transporters. AGRs providing resistance to sulfonamides were detected in the strains of all STs: trimethoprim-resistant dihydrofolate reductase *dfrA17* gene (n = 9) and the sulfonamide-resistant dihydropteroate synthase genes *sul1* (n = 25) and *sul2* (n = 11). Three *A. baumannii* strains of ST2 carried the gene *arr-2*, encoding the NAD(+)-rifampin ADP-ribosyltransferase associated with resistance to rifampin. Moreover, the genetic determinant for the antiseptic-resistant *qacE* gene coding the quaternary ammonium compound efflux SMR-family transporter was detected in the genomes of 25 strains (Figure 3).

### 3.7. The Virulome Identification and Analysis

The virulence factor genes identified via the VFDB online resource were classified into nine virulence factor classes: Adherence (outer membrane protein A, *ompA*, and legionaminic and biosynthesis protein A, *ptmA*), Biofilm formation (AdeFGH efflux pump, *ade*, Biofilm-associated protein, *bap*, Csu pili, *csu*, PNAG Polysaccharide poly-N-acetylglucosamine, *pga*), Enzyme (Phospholipase C, *plc*, Phospholipase D, *plcD*), Immune evasion (Capsule locus, K, and lipopolysaccharide locus, LPS), Iron uptake (Acinetobactin locus, Acb, Heme utilization, HU), Regulation (Quorum sensing, QS, Two-component system, TCS), Serum resistance (Penicillin-binding protein, *pbpG*, dTDP-6-deoxy-L-lyxo-4-hexulose reductase, *rmlD*), Stress adaptation (catalases, *kat*), and Antiphagocytosis (dTDP-4-dehydrorhamnose 3,5-epimerase, *rfbC*, UDP-N-acetylglucosamine 2-epimerase (non-hydrolyzing), *wecB*, UDP-N-acetyl-D-mannosamine dehydrogenase, *wecC*).

A total of 20 types of putative virulence factor genes (VFGs) were predicted in 37 *A. baumannii* genomes. The VFG open reading frames (ORFs) associated with Adherence (n = 39), Biofilm formation (n = 537), Enzyme (n = 139), Immune evasion (n = 964), Iron uptake (n = 989), Regulation (n = 124), Serum resistance (n = 32), Stress adaptation (n = 130), and Antiphagocytosis (n = 7) were identified in the genomes (Figure 4).

The VFGs identified in all 37 *A. baumannii* genomes were the following: *ompA*, *ade*, *csu*, *pga*, *plc*, plcD, Capsula locus, LPS locus, Acb locus, two-component system genes, *pbpG*, and *kat*. At the same time, the *ptmA* gene encoding legionaminic and biosynthesis protein A was detected in two strains of ST2 and ST78 only. The *bap* gene encoding biofilm-associated protein was detected in all strains except two strains of ST1 and four strains of ST19 and ST106. The HU locus was obtained in all strains except one of ST106. The *abaI* inducer and *abaR* receptor genes associated with QS were detected in 30 strains. The *rmlD* gene encoding dTDP-6-deoxy-L-lyxo-4-hexulose reductase provides serum resistance, as well as the *rfbC* gene coding dTDP-4-dehydrorhamnose 3,5-epimerase providing antiphagocytosis, were detected in only one strain of ST106. The *wecB* and *wecC* genes encoding UDP-N-acetylglucosamine 2-epimerase (non-hydrolyzing) and UDP-N-acetyl-D-mannosamine dehydrogenase, respectively, providing antiphagocytosis, were identified in the strains of ST19 only. Despite the vast variability of VFGs in the strains under study, the correlation between disease outcomes—patient death or survival—was not obtained.

Upon analyzing *Acinetobacter* genomes through the VFDB online resource, it was determined that the “*katA* Catalase (*Neisseria*)” gene is present in all strains belonging to ST2. This data prompted further examination of *kat* genes in all studied *Acinetobacter* genomes. These genes include *katA*, which encodes a small-subunit mono-functional catalase; *katE*, the large-subunit mono-functional catalase; *katG*, the catalase-peroxidase; and *katX,* a small protein with a catalase-domain. The reference genome of the *A. baumannii* strain AB5075-UW (CP008706.1) isolated from the tibia/osteomyelitis of the diabetes patient in the USA in 2008 was obtained from GenBank (Table 5).

The *katA* gene with 100% identity compared to the reference (GB accession number AKA32231.1) was detected in three strains of ST1 and two strains of ST19. All strains of ST2 and ST45 carried the *katA* gene with 25–26 single nucleotide polymorphisms (SNPs), and only one SNP was significant, resulting in the single amino acid variation (SAV), g367c/A123P. The strains of ST78 and ST400 carried the *katA* gene with 21 and 23 SNPs, respectively, which caused two SAVs, a211t/T71S and g367c/A123P, in the ST78 strains, as well as g367c/A123P and g649t/V217F in the ST400 strain. The *katA* gene was not detected in strain B-9401 of ST19, while the pseudogene was identified in strain B-9406 of ST400. It should be noted that SAV g367c/A123P was detected in all variants of the *katA* gene in this study.

The sequence of the *katE* gene was matched to the reference (GB accession number AKA32163.1). The 100%-identical genes were detected in stains of ST1 and two strains of ST19. All strains attributed to the ST2 and ST45 carried 35 SNPs in this gene, providing three SAVs: a876c/Q292H; c1304t/A435V; and a1610g/N537S. The strains of ST78 carried the *katE* gene with 30 SNPs compared to the reference, including two nonsynonymous substitutions, g1303a and c1304t, resulting in one SAV A435I. The strains B-9404 of ST106 and B-9406 of ST400 carried 25 and 24 SNPs, respectively, realizing in one (a1820g/N607S) and two (c1304t/A435V and t1939g/L647V) SAVs, respectively. The *katE* gene was missing in strain B-9401 of ST19. Interestingly, that SAV in position 435 was common for *katE* genes in this study, two variants of SAVs were detected: A435V and A435I.

The *katG* gene in 100%-identity (compared to reference AKA33165.1) was detected in the strains of ST1 and ST19. At the same time, all strains attributed to the ST2 and ST45 carried 24 SNPs in this gene, providing four SAVs, namely, c808t/P270S, a1264g/T422A, c1366t/P456S, and g1369a/A457T. Even though the strains of ST78 and ST106 carried different SNPs (27 and 31, respectively) in the *katG* gene, they have four identical SAVs (c808t/P270S; a1264g/T422A; c1366t/P456S; and a1513c/K505Q). The strain B-9406 belonged to ST400 and carried 26 SNPs in the *katG* gene providing two SAVs, namely, c808t/P270S and a1264g/T422A. It is noteworthy that two SAVs (c808t/P270S and a1264g/T422A) were common for mutant *katG* genes in the stains of ST2, ST45, ST78, ST106, and ST400; while the SAV (c1366t/P456S) was frequent in the strains of ST2, ST45, ST78, and ST106; and the SAV (a1513c/K505Q) was prevalent in the strains of ST78 and ST106.

The *katX* gene was detected in 22 of 37 *A. baumannii* strains in the study; meanwhile, two strains of ST1 and three strains of ST19 carried this gene with 100%-identity (compared to reference) AKA31788.1. All the *katX* gene-positive strains carried the same 25 SNPs, resulting in six SAVs (a85g/T29A; g461a/G154D; a694g/T232A; t784a/S262T; a790g/T264A; and t922a-t923c/F308T). The stains attributed to ST78 carried the pseudogene (Table 5).

### 3.8. Phylogenetic Analysis

The phylogenetic relationship between 37 *A. baumannii* genomes determined on the base of the SNPs analysis showed the presence of five main clusters. The first cluster included 24 genome sequences of the strains belonging to ST2 and two strains of ST45; the second cluster included the genomes belonging to ST1 and ST19; while the genomes of the strains belonging to ST78, ST106, and ST400 formed individual clusters each. The phylogenetic analysis also showed that the genomes belonging to ST2 were divided into five sub-clusters, while the genomes of the other STs were combined in a single cluster for a specific type (Figure 5).

## 4. Discussion

The present study characterized 37 *Acinetobacter baumannii* strains isolated from 37 patients of Moscow neurosurgery ICU with meningitis in 2013–2020, which caused 37.8% of deaths. The overall incidence of nosocomial meningitis in this period was 9.5%; the average proportion of *A. baumannii* among all nosocomial meningitis causative agents was 22.1%. [31]. Such a high level of mortality is in agreement with reports about *A. baumannii* as a problematic pathogen to human health due to its high level of drug resistance, increased virulence, and difficulty in treatment, with a high mortality rate (15–73%) because of its multiple drug resistance [32,33].

Seven *A. baumannii* STs were revealed in the study: ST1, ST2, ST19, ST45, ST78, ST106, and ST400. The prevalent ST2 belonging to the pandemic lineage, named “international clone IC2”, was associated in our study with 12 of 14 deaths. ST2 is a well-known dominant type (59%) among the available complete and draft genomes in the GenBank database. This is also consistent with a large number of previous publications that report outbreaks due to IC2s accounting for the bulk of carbapenem-resistant *A. baumannii* (CRAB) outbreaks, whilst IC2 strains do not seem to be the dominant type in South American countries, Tunisia, Tanzania, Poland, and Japan [34]. Other *A. baumannii* strains associated with the death of the patients were ST45, belonging to IC2; and ST400, attributed to the distant phylogenetic group. The last ST was previously rare and described for a nosocomial *A. baumannii* isolate in Germany [35].

Thirteen KLs were identified in our study; the most prevalent was KL9. In *A. baumannii*, the capsular polysaccharide is a major virulence factor that is important for survival in vitro and in vivo. Specific capsule types are proven to overwhelm mammalian defenses in vivo, including K2 [36]. To date, more than 128KL gene clusters (KL types) have been identified at the KL in *A. baumannii* genomes [37]. The major *A. baumannii* outer core locus (OCL) in our study was OCL1, which includes genes for the synthesis of the variable outer core region. This is in agreement with the data of the Kaptive tool that OCL1 is the most common (~74%) amongst isolates attributed to ST1, ST2, ST3, and ST78 [38].

In this study, a majority of the strains were multidrug-resistant MDR, (78%); the rest were extensively drug-resistant XDR, resistant to 6-10 functional groups of antimicrobials. For comparison, the prevalence of XDR *A. baumannii* isolates in Bulgarian university hospitals during the period from 2014 to 2016 was reported as 12.4%; during the period between 2018 and 2019, it increased to 78.1% [39,40]. It was reported recently that intracranial infections caused by drug-resistant Gram-negative bacilli, including MDR, XDR, or even pandrug-resistant (PDR) *A. baumannii*, have very serious consequences [41]. The phenotypic diversity of antibiotic resistance among the strains in our study was very high—23 variants of AMR phenotype were revealed. Carbapenem-resistant strains represented 68%. It is known that CRAB infections are generally nosocomial and frequently occur in the intensive care unit (ICU) for up to 31% around the world with high mortality—greater than 50% in some cases [42]. The carbapenem resistance of the strains in this study was supported by eight *bla*_OXA_-type carbapenemase genes with the prevalence of *bla*_OXA-23_ and *bla*_OXA-80_ genes detected in the strains belonging to ST2 only. This finding is in agreement with reports about *bla*_OXA_-type carbapenemase genes associated with members of GC1 and GC2 [34]. Moreover, the genes of broad-spectrum beta-lactamases and ESBLs (*bla*_TEM-1_, *bla*_CARB-14_, *bla*_CTX-M-124_, *bla*_PER-1_, *bla*_PER-7_, *bla*_GES-11_, *bla*_GES-12_, and *bla*_ADC-25_) were detected in our study. Similar sets of beta-lactamase genes were identified in the studies of other authors [43,44].

Additionally, a total of 47 types of antimicrobial resistance genes (ARGs) providing resistance to aminoglycosides, tetracyclines, macrolides, phenicols, sulfonamides, rifamycin, and antiseptics were identified. Aminoglycoside resistance was associated with *aac*, *aad*, *ant*, *aph*, and *arm* genes. Recently, many reports have appeared showing that carbapenem resistance in *A. baumannii* tends to occur in conjunction with aminoglycosides resistance. The co-production of ArmA 16S rRNA methylase and OXA-23 carbapenemase was determined as the most prevalent mechanism conferring MDR in clinical *A. baumannii* isolates in many countries worldwide, such as Italy, Greece, China, South Korea, India, Yemen, etc. [45]. Among the ARGs providing resistance to other groups of antimicrobials, the *tet* genes were detected in 19/37 strains in this study, which was approximately the same level as the study in Thailand (56.1%); a similar level of prevalence was also obtained for the *arr-2* gene encoding the resistance to rifampin—3/37 strains in our study (9.1%)—in Thailand. Regarding phenicol resistance markers—*cat, cmlA*, and *floR* genes—17/37 strains carried these determinants in our study, which was higher than in the Thailand study (8.1%). As well as the prevalence of AGRs providing resistance to sulfonamides, *sul1* and sul2 genes were detected in 25/37 strains and 11/37 strains, respectively, in our study, compared to 46.1% and 13.6% in Thailand. In contrast, the *msrE* and *mphE* genes providing the macrolide resistance were less common in our study (12/37 strains) than in Thailand (79.6%) [46]. Importantly, three colistin-resistant strains were identified among *A. baumannii* causing meningitis, all of which were attributed to ST2 as in the previously described colistin-resistant isolates collected from the urinary tract, respiratory tract, skin, soft tissue, and blood of patients in some Russian regions in 2013–2014 [47].

Moreover, the genetic determinant for the antiseptic-resistant *qacE* gene was detected in the genomes of 25/37 strains in our study, which was higher than the previously published prevalence of this gene in *A. baumannii* isolates collected from urine, pus, blood, and endotracheal fluid/aspirate in Malaysia (28%). However, the phenotypic resistance of strains in our study was lower (MIC = 4–8 mg/L of benzalkonium chloride; and MIC = 8–64 of chlorhexidine gluconate) than in the Malaysia study (from 12.5 to >50 mg/L and from 25 to >50 mg/L, respectively) [48].

Nine groups of virulence factor genes (VFGs) were identified in *A. baumannii* in this study, including a total of 20 types of VFGs containing the specific numbers of the open reading frames (ORFs): adherence (n=39); biofilm formation (n = 537); enzyme (n = 139); immune evasion (n = 964); iron uptake (n = 989); regulation (n = 124); serum resistance (n = 32); stress adaptation (n = 130); and antiphagocytosis (n = 7). The adherence group was presented by the *ompA* gene detected in all strains. A significantly increasing expression of the *ompA* gene in persisters was recently shown, which proposes *ompA* as a potential target for drug development against *A. baumannii* persisters [49]. Additionally, the *ptmA* gene was detected in two strains of ST2 and ST7; this is in agreement with the opinion that this virulence factor is quite rare [50]. Several genes involved in the biofilm formation of *A. baumannii* were identified in the strains in this study, such as *adeFGH* efflux pump genes, *bap* gene, *csu* locus, and *pga* locus. However, the capacity to form biofilms varied among the strains, with the majority of the isolates forming either moderate (8/37) or weak (17/37) biofilms and only a minority (2/37) forming strong biofilms. This is in agreement with other studies aimed at the evaluation of the distribution of VFGs involved in biofilm formation in multi-drug-resistant *A. baumannii* [51]. The *plc* and *plcD* genes contributing to pathogenesis by aiding in the lysis of host cells to facilitate bacterial invasion were also detected in all studied *A. baumannii* strains. The major differences between the strains were observed in the capsular export region, where the number of genes varied from 11 to 23. These findings were in agreement with recently published data [52]. A number of putative genes associated with acinetobactin biosynthesis were detected in all *A. baumannii* strains in our study and varied from 16 to 21. In contrast, the heme utilization locus was obtained in all strains except one of ST106. Surprisingly, only this strain of ST106 contained the additional gene *rmlD* involved in serum resistance, as well as the *rfbC* gene providing antiphagocytosis activity [53]. Surprisingly, the VFDB online resource revealed “*katA* Catalase (*Neisseria*)” in all *A. baumannii* strains belonging to ST2. A further comprehensive examination of *kat* genes showed that the strains carried the *katA* (35/37), *katE* (36/37), *katG* (37/37), and *katX* (23/37) genes. This finding is consistent with current knowledge of the significance of *katE* and *katG* genes in H_2_O_2_ resistance, as well as the lesser influence of *katA* and *katX* genes on peroxide resistance [54]. It should be noted that putative KatE was different in the strains of ST2 and ST45 (Q292H, A435V, and N537S), ST78 (A435I), ST106 (N607S), and ST400 (A435V and L647V) compared to the KatE of ST1 strains. At the same time, putative KatG varied in the strains of ST2 and ST45 (P270S, T422A, P456S, and A457T), ST78 and ST106 (P270S, T422A, P456S, and K505Q), and ST400 (P270S and T422A) compared to ST1 and ST19. This observation demonstrates the ongoing evolution of these virulence factor determinants. The wide variety of the virulence factors identified in the *Acinetobacter* strains that caused meningitis in neuro-intensive care patients indicates the plasticity of the pathogen genome. This is consistent with the previously described diverse interaction between different pathogenicity factors, such as iron acquisition, cell adherence, cell motility, biofilm formation, and antibiotic susceptibility [55].

In conclusion, this study provides a detailed account of the genetic determinants associated with multidrug resistance and virulence in clinically significant *A. baumannii* isolates from the patients of a Moscow neurosurgery ICU with meningitis. The sequence type ST2 was the prevalent genetic line in this cohort. Numerous variants of antimicrobial resistance genes and virulence gene sets were revealed in the genomes of meningitis agents. At the same time, the correlation between STs and their molecular–genetic characteristics was obtained. This observation could form the basis for the future design of novel diagnostic and treatment strategies. We propose to continue this study by focusing on the further investigation of antimicrobial resistance islands, plasmid compositions, the localization of the genetic mobile elements, etc.

## Figures and Tables

**Figure 1 microorganisms-11-02020-f001:**
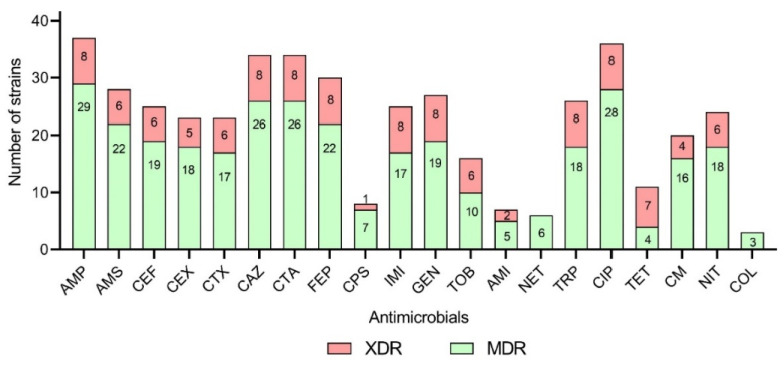
The number of *A. baumannii* strains of MDR and XDR phenotypes, resistant to antibiotics: AMP, ampicillin; AMS, ampicillin–sulbactam; CEF, cefuroxime; CEX, cefoxitin; CTX, cefotaxime; CAZ, ceftazidime; CTA, ceftriaxone; FEP, cefepime; CPS, cefoperazone/sulbactam; IMI, imipenem; GEN, gentamicin; TOB, tobramycin; AMI, amikacin; NET, netilmicin; TRP, trimethoprim–sulfamethoxazole; CIP, ciprofloxacin; TET, tetracycline; CM, chloramphenicol; NIT, nitrofurantoin; COL, colistin.

**Figure 2 microorganisms-11-02020-f002:**
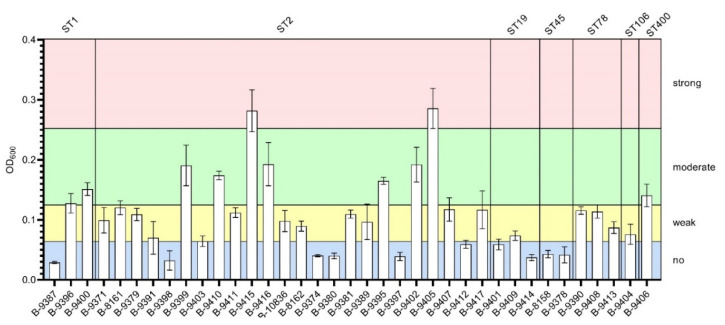
Biofilm formation ability for *A. baumannii* strains causing meningitis in Moscow neuro-ICU in 2013–2020.

**Figure 3 microorganisms-11-02020-f003:**
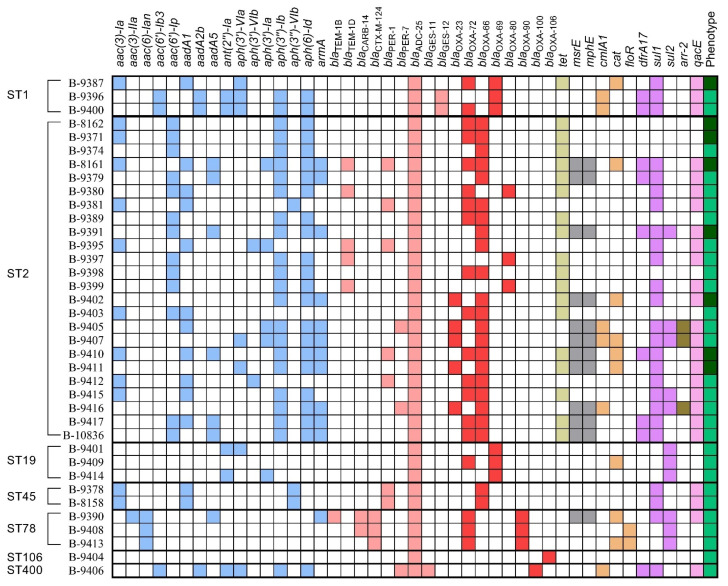
Resistomes of 37 *A. baumannii* strains causing meningitis in Moscow Neurological ICU in 2013–2020. ARGs of different groups are presented in specific colors. MDR phenotype is designated by the green color, and XDR phenotype is designated by the dark green color.

**Figure 4 microorganisms-11-02020-f004:**
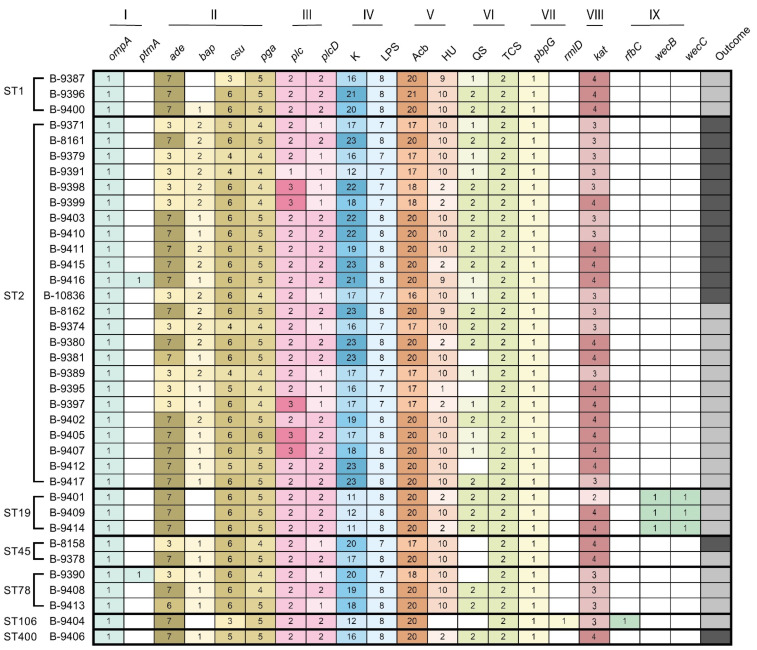
Virulomes of 37 *A. baumannii* strains caused meningitis in Moscow Neurological ICU in 2013–2020. Virulence genes of different groups are presented in specific colors: I, Adherence; II, Biofilm formation; III, Enzyme; IV, Immune evasion; V, Iron uptake; VI, Regulation; VII, Serum resistance; VIII, Stress adaptation; IX, Antiphagocytosis. The color intensity reflects the number of predicted ORFs. The survival outcome is designated by the gray color, death outcome is designated by the dark gray color.

**Figure 5 microorganisms-11-02020-f005:**
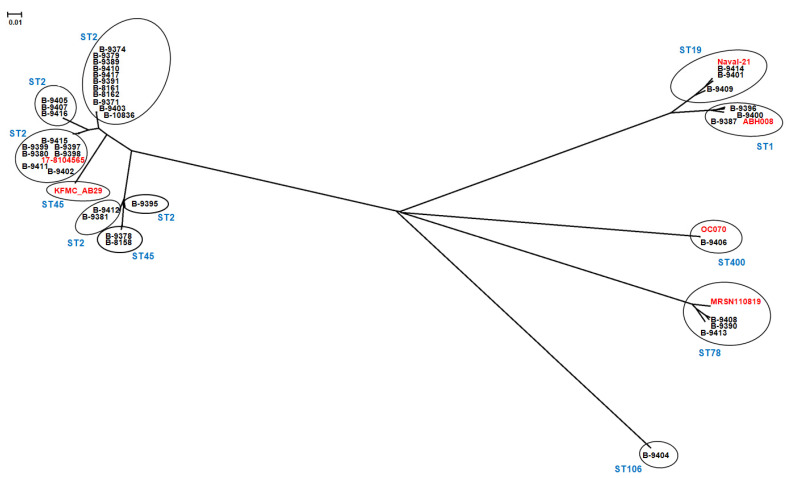
Neighbor-joining phylogenetic tree based on 58,859 core SNPs, showing the phylogenetic relationships between 37 genomes of *A. baumannii* strains causing meningitis in this study (black color) and 6 *A. baumannii* genomes accepted as references from the NCBI database (red color): strain ST1 isolate A076 (KT317084.1), Sweden; strain ST2 isolate 17-8104565 (CANDXI000000000.1), United Kingdom; strain ST19 isolate Naval-21 (AMSY00000000.1), USA; strain ST45 isolate KFMC_AB29 (JALGJO000000000.1), Kingdom of Saudi Arabia; strain ST78 isolate 4190 (KT266827.1), Australia; strain ST400 isolate OC070 (CP087298.1), Germany. The scale bar represents a 0.01 base substitution per site.

**Table 1 microorganisms-11-02020-t001:** Patient information.

Case	Gender, Age	Date	Patient’s Diagnosis	Infections	Stay in the ICU, Days	Outcome
#1	M, 16	15 January 2013	Medulloblastoma of the fourth ventricle	MG, RTI, SSI, GD	186	Survival
#2	M, 46	27 February 2013	Anaplastic astrocytoma	MG, RTI, UTI	24	Died
#3	M, 50	4 February 2014	Cholesteatoma	MG, BI, GD	184	Survival
#4	F, 65	25 June 2014	Suprasellar craniopharyngioma	MG	31	Died
#5	F, 24	5 November 2014	Anterior cerebral artery aneurysm	MG	62	Survival
#6	F, 52	19 January 2015	Craniofacial tumor	MG, SSI, UTI	34	Died
#7	M, 48	4 February 2015	Pilocytic astrocytoma	MG, RTI	43	Survival
#8	M, 33	29 September 2015	Traumatic brain injury	MG, RTI, SSI	45	Survival
#9	F, 2	13 November 2015	Cerebellar astrocytoma	MG, RTI, SSI	62	Survival
#10	M, 57	7 December 2015	Giant cell tumor of the middle cranial fosses	MG	15	Survival
#11	M, 44	9 December 2015	Anterior cerebral artery aneurysm	MG, RTI	8	Died
#12	M, 43	18 May 2016	Traumatic brain injury	MG	15	Survival
#13	M, 43	15 June 2016	Cerebellum tumor	MG, RTI, SSI	63	Died
#14	M, 12	16 June 2016	Giant craniopharyngioma	MG	62	Survival
#15	M, 28	1 August 2016	Atypical parasagittal meningioma	MG, SSI	49	Survival
#16	F, 60	2 August 2016	Giant olfactory meningioma	MG	229	Died
#17	M, 75	16 August 2016	Ependymoma	MG, RTI, UTI	136	Died
#18	M, 35	12 September 2016	Glioblastoma	MG	98	Died
#19	F, 57	10 October 2016	Traumatic brain injury	MG, RTI, UTI	14	Survival
#20	M, 67	29 November 2016	Sub-ependymoma	MG	31	Survival
#21	M, 61	4 September 2017	Craniopharyngioma	MG	133	Survival
#22	F, 61	12 December 2017	Hemangiopericytoma	MG	24	Survival
#23	F, 36	15 January 2018	Giant meningioma	MG, RTI, SSI	65	Died
#24	M, 55	21 February 2018	Hemangioblastoma	MG	46	Survival
#25	M, 57	20 June 2018	Gunshot wound head trauma	MG, RTI	181	Survival
#26	F, 58	19 September 2018	Glioma	MG	52	Died
#27	M, 43	1 October 2018	Hemangioblastoma of the cerebellum	MG, RTI	53	Survival
#28	M, 53	23 October 2018	Traumatic brain injury	MG, RTI	58	Survival
#29	M, 53	12 November 2018	Traumatic brain injury	MG, RTI, UTI, GD	103	Survival
#30	M, 61	3 December 2018	Post-hemorrhagic hydrocephalus	MG, RTI, GD	19	Died
#31	M, 64	15 May 2019	Skull base chordoma	MG, RTI	25	Died
#32	F, 25	20 May 2019	Traumatic brain injury	MG, RTI	52	Survival
#33	M, 26	18 July 2019	Traumatic brain injury	MG, RTI	45	Survival
#34	M, 51	6 August 2019	Ruptured middle cerebral artery aneurysms	MG	26	Survival
#35	M, 47	27 January 2020	Giant skull base chordoma	MG	89	Died
#36	F, 52	28 August 2020	Syringomyelia of the cervical spinal cord	MG, RTI	71	Died
#37	F, 45	8 September 2020	Hemangiopericytoma	MG, GD	69	Survival

Note: M, Male; F, Female; MG, Meningitis; RTI, Respiratory Tract Infection; SSI, Surgical Site Infections; BI, Blood Infection; GD, Gastrointestinal Dysfunction; UTI, Urinary Tract Infection.

**Table 2 microorganisms-11-02020-t002:** Characteristics of *A. baumannii* strains caused meningitis in neuro-ICU in 2013–2020.

Case	Strain	ST	KL	OCL	AMR Phenotype	AM gr.	ARP
#1	B-8162	2	9	1	AG, CAR, FQN, CEF, SUL, PEN, PEN-IN, TET, NIT	9	XDR
#2	B-9371	2	9	1	AG, CAR, FQN, CEF, SUL, PEN, PEN-IN, NIT	8	XDR
#3	B-9374	2	9	1	AG, CAR, FQN, CEF, PEN, TET, CM, NIT	8	MDR
#4	B-8161	2	9	1	AG, CAR, FQN, CEF, SUL, PEN, PEN-IN, TET, CM, NIT	10	XDR
#5	B-9378	45	77	1	AG, CAR, FQN, CEF, PEN, PEN-IN, CM, NIT	8	MDR
#6	B-9379	2	9	1	AG, CAR, FQN, CEF, SUL, PEN, TET, CM, NIT	9	MDR
#7	B-9380	2	2	1	AG, FQN, CEF, PEN, PEN-IN, TET, CM, NIT	8	MDR
#8	B-9381	2	9	1	AG, CAR, FQN, CEF, PEN, PEN-IN, CM, NIT	8	MDR
#9	B-9387	1	17	1	AG, CAR, FQN, CEF, PEN, TET, CM, NIT	8	XDR
#10	B-9389	2	9	1	AG, CAR, FQN, CEF, SUL, PEN, CM, NIT	8	MDR
#11	B-10836	2	9	1	AG, CAR, FQN, CEF, PEN, PEN-IN, CM, NIT	8	MDR
#12	B-9390	78	49	1	AG, CAR, FQN, CEF, SUL, PEN, PEN-IN, CM, NIT	9	MDR
#13	B-9391	2	9	1	AG, CAR, FQN, CEF, SUL, PEN, TET, CM, NIT	9	XDR
#14	B-9395	2	165	1	AG, FQN, CEF, PEN, PEN-IN, CM, NIT	7	MDR
#15	B-9397	2	213	1	AG, FQN, CEF, PEN, PEN-IN, POL, NIT	7	MDR
#16	B-9398	2	2	1	AG, FQN, CEF, SUL, PEN, PEN-IN, POL, NIT	8	MDR
#17	B-8158	45	9	1	AG, FQN, CEF, PEN, POL, CM	6	MDR
#18	B-9399	2	213	1	AG, FQN, CEF, PEN, PEN-IN, TET, CM, NIT	8	MDR
#19	B-9396	1	4	1	AG, FQN, CEF, SUL, PEN, PEN-IN, CM, NIT	8	MDR
#20	B-9400	1	4	1	AG, FQN, CEF, SUL, PEN, PEN-IN, CM, NIT	8	MDR
#21	B-9401	19	91	5	AG, FQN, CEF, SUL, PEN, PEN-IN, CM, NIT	8	MDR
#22	B-9402	2	3	1	AG, CAR, FQN, CEF, SUL, PEN, PEN-IN, TET, CM, NIT	10	XDR
#23	B-9403	2	9	1	AG, CAR, FQN, CEF, PEN, CM, NIT	7	MDR
#24	B-9404	106	104	6	CEF, SUL, PEN, PEN-IN, CM, NIT	6	MDR
#25	B-9405	2	235	1	AG, CAR, FQN, CEF, SUL, PEN, PEN-IN	7	MDR
#26	B-9406	400	15	1	AG, FQN, CEF, SUL, PEN-IN	6	MDR
#27	B-9407	2	235	1	AG, CAR, FQN, CEF, SUL, PEN, PEN-IN	7	MDR
#28	B-9408	78	3	1	CAR, FQN, CEF, SUL, PEN, PEN-IN	6	MDR
#29	B-9409	19	91	5	CAR, FQN, CEF, SUL, PEN, PEN-IN	6	MDR
#30	B-9410	2	9	1	AG, CAR, FQN, CEF, SUL, PEN, PEN-IN, TET	8	XDR
#31	B-9411	2	3	1	AG, CAR, FQN, CEF, SUL, PEN, PEN-IN, TET	8	XDR
#32	B-9412	2	9	1	CAR, FQN, CEF, PEN, PEN-IN, NIT	6	MDR
#33	B-9413	78	15	1	CAR, FQN, CEF, SUL, PEN, PEN-IN	6	MDR
#34	B-9414	19	91	5	AG, FQN, CEF, SUL, PEN, PEN-IN	6	MDR
#35	B-9415	2	2	1	AG, CAR, FQN, CEF, SUL PEN,	6	MDR
#36	B-9416	2	49	1	AG, CAR, FQN, CEF, SUL, PEN, POL	7	MDR
#37	B-9417	2	9	1	AG, CAR, FQN, CEF, SUL, PEN, PEN-IN	7	MDR

Note: ST, sequence type; CL, capsular locus type; OCL, lipooligosaccharide outer core locus type; AMR, antimicrobial resistance phenotype; AM gr., number of antimicrobial groups (resistance); ARP, antimicrobial resistance patterns [21]; AG, aminoglycosides; CAR, carbapenems; FQN, fluoroquinolones; CEF, cephalosporins; SUL, sulfonamides; PEN, penicillin; PEN-IN, penicillin/beta-lactam inhibitors; TET, tetracyclines; CM, phenicols (chloramphenicol); POL, polymyxins; NIT, nitrofurans; MDR, multidrug resistant; XDR, extensively drug resistant.

**Table 3 microorganisms-11-02020-t003:** Susceptibility of *A. baumannii* planktonic cells to antiseptics.

Strain	MIC, mg/L		Strain	MIC, mg/L	
	CHD	BZK		CHD	BZK
B-9387	32	8	B-9389	32	8
B-9396	64	8	B-9395	32	8
B-9400	64	8	B-9397	32	8
B-9371	32	8	B-9402	32	4
B-8161	32	8	B-9405	16	4
B-9379	32	8	B-9407	32	8
B-9391	32	8	B-9412	32	8
B-9398	32	8	B-9417	32	8
B-9399	32	8	B-9401	32	8
B-9403	32	8	B-9409	16	8
B-9410	32	8	B-9414	16	4
B-9411	16	4	B-8158	16	8
B-9415	8	4	B-9378	16	8
B-9416	32	8	B-9390	16	4
B-10836	32	8	B-9408	32	8
B-8162	32	8	B-9413	16	4
B-9374	32	8	B-9404	16	4
B-9380	32	8	B-9406	32	8
B-9381	32	8			

Note: CHD, chlorhexidine digluconate; BZK, benzalkonium chloride.

**Table 4 microorganisms-11-02020-t004:** Whole-genome characteristics for *A. baumannii* strains causing meningitis in Moscow neuro-ICU in 2013–2020.

Strain	GenBank ID	GC Content, %	Genes (Total)	CDSs (Total)	Genes (Coding)	Genes (RNA)	rRNA Genes (5S, 16S, 23S)	tRNAs	ncRNAs	Pseudo Genes
B-8162	JAROBQ000000000	39.8	3929	3858	3801	71	1, 1, 1	64	4	57
B-9371	JASKJB000000000	42.4	3722	3650	3600	72	0, 2, 3	63	4	50
B-9374	JASKJA000000000	42.4	3873	3802	3745	71	0, 2, 1	64	4	57
B-8161	JAROBM000000000	39.2	3848	3778	3723	70	1, 1, 1	63	4	55
B-9378	JAROBU000000000	39.6	3707	3636	3576	71	1, 1, 1	64	4	60
B-9379	JASKIZ000000000	42.4	3704	3632	3580	72	1, 2, 2	63	4	52
B-9380	JAROBS000000000	39.9	3951	3880	3829	71	1, 1, 1	64	4	51
B-9381	JAROBO000000000	39.3	3867	3797	3745	70	1, 1, 1	63	4	52
B-9387	JAROBG000000000	39.2	3877	3807	3737	70	1, 1, 1	63	4	70
B-9389	JASKIY000000000	42.2	3596	3524	3475	72	0, 2, 3	63	4	49
B-9390	JASKIX000000000	42.4	3890	3820	3764	70	1, 1, 1	63	4	56
B-9391	JASKIW000000000	42.1	3679	3608	3555	71	0, 2, 2	63	4	53
B-9395	JASKIV000000000	42.9	3887	3816	3758	71	0, 2, 1	64	4	58
B-9397	JASKIU000000000	42.2	3795	3725	3676	70	0, 1, 1	64	4	49
B-9398	JAROBR000000000	39.8	3965	3894	3840	71	1, 1, 1	64	4	54
B-8158	JAROBT000000000	40.3	3735	3664	3605	71	1, 1, 1	64	4	59
B-9399	JASKIT000000000	42.0	3810	3741	3692	69	0, 1, 1	63	4	49
B-9396	JAROBB000000000	40.1	3855	3785	3720	70	1, 1, 1	63	4	65
B-9400	JAROBI000000000	39.8	3879	3809	3744	70	1, 1, 1	63	4	65
B-9401	JASKIR000000000	40.6	3624	3553	3496	71	1, 1, 1	63	5	57
B-9402	JASKIQ000000000	39.3	3797	3726	3664	71	1, 1, 1	64	5	62
B-9403	JASKIP000000000	39.2	4019	3949	3891	70	1, 1, 1	63	5	58
B-9404	JASKIO000000000	39.7	3821	3752	3644	69	0, 1, 1	63	5	108
B-9405	JASKIN000000000	39.2	3957	3887	3830	70	1, 1, 1	70	5	57
B-9406	JASKIM000000000	39.3	3803	3733	3675	70	1, 1, 1	63	4	58
B-9407	JASKIL000000000	39.3	3956	3886	3830	70	1, 1, 1	63	4	56
B-9408	JASKIK000000000	39.9	3804	3733	3682	71	1, 1, 1	64	4	51
B-9409	JASKIJ000000000	39.5	3845	3775	3716	70	1, 1, 1	62	5	59
B-9410	JASKII000000000	39.4	3853	3783	3728	70	1, 1, 1	63	4	55
B-9411	JASKIH000000000	38.6	3933	3862	3800	71	1, 1, 1	64	4	62
B-9412	JASKIG000000000	39.1	3654	3584	3532	70	1, 1, 1	63	4	52
B-9413	JASKIF000000000	39.4	3745	3674	3626	71	1, 1, 1	64	4	48
B-9414	JASKIE000000000	39.4	3665	3593	3538	72	1, 1, 1	64	5	55
B-9415	JASKID000000000	39.4	3939	3868	3813	71	1, 1, 1	64	4	55
B-9416	JASKIC000000000	39.4	3928	3859	3797	69	0, 1, 1	63	4	62
B-9417	JASKIB000000000	39.1	3743	3673	3622	70	1, 1, 1	63	4	51

**Table 5 microorganisms-11-02020-t005:** Analysis of the *kat* genes in the genomes of *A. baumannii* strains causing meningitis.

Strain	ST	*katA* (AKA32231.1)			*katE* (AKA32163.1)			*katG* (AKA33165.1)			*katX* (AKA31788.1)		
		Ident., %	SNP	SAV	Ident., %	SNP	SAV	Ident., %	SNP	SAV	Ident., %	SNP	SAV
B-9387	1	100	0	0	100	0	0	100	0	0	97.85	25	6
B-9396	1	100	0	0	100	0	0	100	0	0	100	0	0
B-9400	1	100	0	0	100	0	0	100	0	0	100	0	0
B-9371	2	97.65	25	1	98.38	35	3	98.89	24	4	-	-	-
B-8161	2	97.65	25	1	98.38	35	3	98.89	24	4	-	-	-
B-9379	2	97.65	25	1	98.38	35	3	98.89	24	4	-	-	-
B-9391	2	97.65	25	1	98.38	35	3	98.89	24	4	-	-	-
B-9398	2	97.65	25	1	98.38	35	3	98.89	24	4	-	-	-
B-9399	2	97,65	25	1	98.38	35	3	98.89	24	4	97.85	25	6
B-9403	2	97.65	25	1	98.38	35	3	98.89	24	4	-	-	-
B-9410	2	97.56	26	1	98.38	35	3	98.89	24	4	-	-	-
B-9411	2	97.65	25	1	98.38	35	3	98.89	24	4	97.85	25	6
B-9415	2	97.65	25	1	98.38	35	3	98.89	24	4	97.85	25	6
B-9416	2	97.65	25	1	98.38	35	3	98.89	24	4	97.85	25	6
B-10836	2	97.65	25	1	98.38	35	3	98.89	24	4	-	-	-
B-8162	2	97.65	25	1	98.38	35	3	98.89	24	4	-	-	-
B-9374	2	97.65	25	1	98.38	35	3	98.89	24	4	-	-	-
B-9380	2	97.65	25	1	98.38	35	3	98.89	24	4	97.85	25	6
B-9381	2	97.65	25	1	98.38	35	3	98.89	24	4	97.85	25	6
B-9389	2	97.65	25	1	98.38	35	3	98.89	24	4	-	-	-
B-9395	2	97.65	25	1	98.38	35	3	98.89	24	4	97.85	25	6
B-9397	2	97.65	25	1	98.38	35	3	98.89	24	4	97.85	25	6
B-9402	2	97.65	25	1	98.38	35	3	98.89	24	4	97.85	25	6
B-9405	2	97.65	25	1	98.38	35	3	98.89	24	4	97.85	25	6
B-9407	2	97.65	25	1	98.38	35	3	98.89	24	4	97.85	25	6
B-9412	2	97.65	25	1	98.38	35	3	98.89	24	4	97.85	25	6
B-9417	2	97.65	25	1	98.38	35	3	98.89	24	4	-	-	-
B-9401	19	-	-	-	-	-	-	100	0	0	100	0	0
B-9409	19	100	0	0	100	0	0	100	0	0	100	0	0
B-9414	19	100	0	0	100	0	0	100	0	0	100	0	0
B-8158	45	97.65	25	1	98.38	35	3	98.89	24	4	97.85	25	6
B-9378	45	97.65	25	1	98.36	35	3	98.89	24	4	97.85	25	6
B-9390	78	98.03	21	2	98.6	30	1	98.75	27	4	pseudo	-	-
B-9408	78	98.03	21	2	98.6	30	1	98.75	27	4	pseudo	-	-
B-9413	78	98.03	21	2	98.6	30	1	98.75	27	4	pseudo	-	-
B-9404	106	pseudo	-	-	98.83	25	1	98.56	31	4	97.85	25	6
B-9406	400	97.84	23	2	98.88	24	2	98.79	26	2	97.85	25	6

Note: ST, sequence type; Ident., identity to the reference gene sequence obtained from GenBank; SNP, single nucleotide polymorphism; SAV, single amino acid variation; “-“, no match; pseudo, pseudogene.

## Data Availability

The following whole-genome sequences were deposited in the GenBank database: JAROBQ000000000, JASKJB000000000, JASKJA000000000, JAROBM000000000, JAROBU000000000, JASKIZ000000000, JAROBS000000000, JAROBO000000000, JAROBG000000000, JASKIY000000000, JASKIX000000000, JASKIW000000000, JASKIV000000000, JASKIU000000000, JAROBR000000000, JAROBT000000000, JASKIT000000000, JAROBB000000000, JAROBI000000000, JASKIR000000000, JASKIQ000000000, JASKIP000000000, JASKIO000000000, JASKIN000000000, JASKIM000000000, JASKIL000000000, JASKIK000000000, JASKIJ000000000, JASKII000000000, JASKIH000000000, JASKIG000000000, JASKIF000000000, JASKIE000000000, JASKID000000000, JASKIC000000000, JASKIB000000000.

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
