# Peer review of "Multidrug-Resistant and Extensively Drug-Resistant Acinetobacter baumannii Causing Nosocomial Meningitis in the Neurological Intensive Care Unit"

_microorganisms, 2023, doi:10.3390/microorganisms11082020_

Round 1

Reviewer 1 Report

In their manuscript entitled "Multidrug-Resistant and Extensively Drug-Resistant Acinetobacter baumannii Causing Nosocomial Meningitis in the Neurological Intensive Care Unit", Fursova and co-authors analyzed 37 A. baumannii isolates collected from 37 patients with meningitis hospitalyzed at a Moscow Neurological intensive care unit from 2013 to 2020. The authors have performed the isolates whole genomes sequencing,  and describe the isolates´ antimicrobial susceptibility, biofilm formation ability , selected virulence factors and genetic determinants of antimicrobials resistance in the whole-genome sequences. 

The manuscript is well organized and the conclusions taken aresupported by the results. There are however some issues that require the authors action:

Abstract: the abstract needs to be reorganized.  Suggestion: After the first sentence, introduce the general characterization of the isolates and their origin. Then, mention the studies performed with the and finally end with some major conclusions.

In page 3, the sentence "The isolates were categorized as resistant (R), non-susceptible to ≥1 agent in <3 antimicrobial categories; multidrug-resistant (MDR), non-susceptible to ≥1 agent in ≥3 antimicrobial categories; extensively drug-resistant (XDR), non-susceptible to ≥1 agent in all but ≤2 categories; according to the criteria proposed by Magiorakos et al., 2012." is confusing. Please rewrite. The reference should be as follows: Magiorakos et al. [21].

In page 3, "Sigma-Aldrich, Saint Louise, MO, USA" should be "Sigma-Aldrich, Saint Louis, MO, USA"

page 6: the reference "...Magiorakos et al. (2012) criterium [1]." is wrong. Please correct as follows: "... Magiorakos et al. criterium [21]."

Page 13: "...Capsule locus..."

Table 5: The "," in identity should be"." 

4th line: B-9371 2 97.65 25 1 98.38 35 3 98.89

Correct all the other lines.

Figure 5: what is the meaning of the 0.01 in the scale? Please indicate in the legend.

In page 14 a significant part of th emanuscript is dedicated to the analysis o kat genes and no rationale is given for such a detailed analysis. Please provide a justification for this analysis.

English language is fine, minor corrections are needed.

Author Response

We are grateful to Reviewer 1 for carefully reading our manuscript and questions. We can answer the questions as follows:

Q1. Abstract: the abstract needs to be reorganized. Suggestion: After the first sentence, introduce the general characterization of the isolates and their origin. Then, mention the studies performed with the and finally end with some major conclusions.

Answer 1. According to Reviewer’s suggestion, we modified the Abstract: “Acinetobacter baumannii is one of the significant healthcare-associated meningitis agents characterized by multidrug resistance and a high mortality risk. Thirty-seven A. baumannii strains were isolated from 37 patients of Moscow neuro-ICU with meningitis in 2013-2020. The death rate was 37.8 %. Strain susceptibility to antimicrobials was determined on the Vitek-2 instrument…”.

Q2. In page 3, the sentence "The isolates were categorized as resistant (R), non-susceptible to ≥1 agent in <3 antimicrobial categories; multidrug-resistant (MDR), non-susceptible to ≥1 agent in ≥3 antimicrobial categories; extensively drug-resistant (XDR), non-susceptible to ≥1 agent in all but ≤2 categories; according to the criteria proposed by Magiorakos et al., 2012." is confusing. Please rewrite. The reference should be as follows: Magiorakos et al. [21].

Answer 2. We rewrote the phrase: “The isolates were categorized as multidrug-resistant (MDR), non-susceptible to ≥1 agent in ≥3 antimicrobial groups; extensively drug-resistant (XDR), non-susceptible to ≥1 agent in all but ≤2 groups; according to the criteria proposed by Magiorakos et al. [21].”

Q3. In page 3, "Sigma-Aldrich, Saint Louise, MO, USA" should be "Sigma-Aldrich, Saint Louis, MO, USA"

Answer 3. It was corrected.

Q4. page 6: the reference "...Magiorakos et al. (2012) criterium [1]." is wrong. Please correct as follows: "... Magiorakos et al. criterium [21]."

Answer 4. It was corrected.

Q5. Page 13: "...Capsule locus..."

Answer 5. It was corrected.

Q6. Table 5: The "," in identity should be"."  (4th line: B-9371 2 97.65 25 1 98.38 35 3 98.89) Correct all the other lines.

Answer 6. All lines were corrected.

Q7. Figure 5: what is the meaning of the 0.01 in the scale? Please indicate in the legend.

Answer 7. The phrase “The scale bar represents a 0.01 base substitution per site.” was added in the legend of Figure 5.

Q8. In page 14 a significant part of the manuscript is dedicated to the analysis of kat genes and no rationale is given for such a detailed analysis. Please provide a justification for this analysis.

Answer 8. According to the Reviewer’s suggestion, we provided the following justification: ”Upon analyzing Acinetobacter genomes through the VFDB online resource, it was determined that the “katA Catalase (Neisseria)” gene is present in all strains belonging to ST2. This data prompted further examination of kat genes in all studied Acinetobacter genomes. These genes include katA, which encodes a small-subunit mono-functional catalase; katE, the large-subunit mono-functional catalase; katG, the catalase-peroxidase; and katX, a small protein with a catalase-domain.”

The corrected file is in the attachment. (Correction.docx)

Reviewer 2 Report

Very impressive work on the characterization of a number of strains of A. baumannii isolated from ICU patients.

Author Response

We are grateful to Reviewer 2 for the positive assessment of our manuscript.

The corrected file is in the attachment. (Correction.docx)

Reviewer 3 Report

The manuscript presents a detailed analysis of A. baumannii isolated from samples from one clinic for seven years. The study is comprehensive and clear. It can serve as an example for writing this type of article. Minor issues that need to be corrected: the Material and Methods chapter does not indicate the method of presenting the results or the type of study in question (descriptive, analytical...). There is no explanation why the occurrence of PDR (Pan-Drug Resistant) strains of A. baumannii is not considered. The discussion is too extensive and mainly represents a repetition of the results. The overall impression is good.

A grammar check is needed.

Author Response

We are grateful to Reviewer 3 for the positive assessment of our manuscript. We can answer the questions as follows:

Q1. the Material and Methods chapter does not indicate the method of presenting the results or the type of study in question (descriptive, analytical...).

Answer 1. The Material and Methods section was completed: “2.1. Research, Bioethical Requirements, and Patients. This study was a retrospective cohort observation analytical research.”

Q2. There is no explanation why the occurrence of PDR (Pan-Drug Resistant) strains of A. baumannii is not considered.

Answer 2. Accordingly to Magiorakos et al. 2012: “The PDR means resistant to all antimicrobial agents.” We did not obtain such strains in our study.

Q3. The discussion is too extensive and mainly represents a repetition of the results.

Answer 3. The Discussion section was corrected.

The corrected file is in the attachment. (Correction.docx)

Round 2

Reviewer 1 Report

The authors have revised the  manuscript and the criticisms raised were adequately solved. No further criticisms.